# Impact of the COVID-19 Lockdown on Physical Activity Levels and Health Parameters in Young Adults with Cancer

Mónica Castellanos-Montealegre [1], Fernando Rivera-Theruel [2], Virginia García-Coll [1,*], Natalia Rioja-Collado [1], Lucía Gil-Herrero [3], Sara López-Tarruella [4,5,6], María Montealegre Sanz [7], Sara Cerezo González [8], Antonio Fernández Aramburo [9], Ana Ruiz-Casado [10], Rebecca Laundos [2] and Soraya Casla-Barrio [11]

1. Department of Science of Physical Activity and Sport, Castilla-La Mancha University Toledo Av de Carlos III, n 21, 45004 Toledo, Spain; monica.castellanos@uclm.es (M.C.-M.); natalia.rioja@uclm.es (N.R.-C.)
2. Toronto Rehabilitation Institute Rumsey Centre Cardiac Rehabilitation, University Health Network, Toronto, ON M4G 2V6, Canada; fernando.riveratheurel@uhn.ca (F.R.-T.); rebecca.laundos@uhn.ca (R.L.)
3. Spanish Cancer Association, Av Federico Rubio y Galí, n 84, 28040 Madrid, Spain; lucia.gil@contraelcancer.es
4. Medical Oncology Service, Hospital General Universitario Gregorio Marañón, Instituto de Investigación Sanitaria Gregorio Marañón (IiSGM), 28007 Madrid, Spain; sara.lopeztarruella@salud.madrid.org
5. CiberOnc, Universidad Complutense, 28040 Madrid, Spain
6. GEICAM, 28703 Madrid, Spain
7. Clínico San Carlos Hospital, 28040 Madrid, Spain; maria.montealegre@salud.madrid.org
8. La Mancha Centro General Hospital, 13600 Alcázar de San Juan, Spain; scerezo@sescam.jccm.es
9. Albacete University Hospital, 02008 Albacete, Spain; afernandeza@sescam.jccm.es
10. HU Puerta de Hierro Majadahonda, IDIPHISA, 28222 Madrid, Spain; arcasado@salud.madrid.org
11. Exercise Oncology Unit, Exercise and Cancer, 28009 Madrid, Spain; soraya@ejercicioycancer.es
* Correspondence: virginia.garcia@uclm.es

**Abstract:** The lockdown of the COVID-19 pandemic impacted physical activity (PA) levels around the world, affecting health parameters in young adults with cancer (YAC). To our knowledge, there is no evidence of the impact of the lockdown on the Spanish YAC. To analyse the changes in PA levels before, during, and after the lockdown of the YAC and its impact on health metrics in Spain, in this study, we utilized a self-reported web survey. PA levels decreased during the lockdown, and a significant increase in PA was observed after the lockdown. Moderate PA had the largest reduction (49%). Significant increases in moderate PA were noted after the lockdown (85.2%). Participants self-reported more than 9 h of sitting per day. HQoL and fatigue levels were significantly worse during the lockdown. The impact of the COVID-19 pandemic in this cohort of Spanish YAC showed a decrease in PA levels during the lockdown, affecting sedentarism, fatigue and HQoL. After lockdown, PA levels partially recovered, while HQoL and fatigue levels remained altered. This may have long-term physical effects such as cardiovascular comorbidities associated with sedentarism and psychosocial effects. It is necessary to implement strategies such as cardio-oncology rehabilitation (CORE), an intervention that can be delivered online, potentially improving participants' health behaviours and outcomes.

**Keywords:** young adults; cancer; physical activity; COVID-19 lockdown; sedentary time; quality of life; fatigue

## 1. Introduction

In Europe, about 252,200 young adults were diagnosed with cancer in 2020; in Spain, 16,000 new cases were reported [1]. In 2019, young adult cancer (YAC) contributed 23.5 million disability-adjusted life years (DALYs) to the global burden of disease, of which 97.3% came from years of life lost [2]. Current evidence shows the impact of improving lifestyle-related risk factors to reduce this burden [3]. Physical inactivity is one of the main risk factors contributing to DALYs in different cancers including breast [4] and colon cancer [5].

It is suggested that cancer in young adults could differ molecularly from other age groups, possibly suggesting differences in both etiologic factors and the efficacy of cancer treatment [6]. Due to the exponential growth in the number of YAC survivors as a result of improved screening and therapeutic strategies, oncological treatments in this population [7] have been related to a higher risk of long-lasting side effects and toxicities [8]. Fatigue is one of the most common side effects of cancer therapies, especially during and after chemotherapy or radiotherapy administration [9]. It has been found that fatigue can persist long after the end of treatment, ref. [10] impairing health-related quality of life (HQoL). Numerous studies in the YAC population have demonstrated that a physical activity (PA) intervention can prevent and mitigate cancer-related fatigue, ref. [11,12] with the potential to improve HQoL [13].

Higher levels of PA (>150 per week as recommended in the American Cancer Society and Sports Medicine guidelines) in YAC patients have been shown to be effective at mitigating some cardiovascular sequelae from cancer therapies, such as the reduction in cardiorespiratory fitness capacity (peak VO2), ref. [14] and the improvement of the long-term risk of cardiovascular disease (CVD), ref. [15] which represents an important concern in cancer survivors [16]. Higher PA levels before, during and after cancer diagnosis have been associated with a reduction in CVD- and cancer-related morbidity and mortality risk [17]. However, prior studies reported that 40% of the YAC population do not meet PA guidelines before or after cancer diagnosis and only 5% participate in PA programs for cancer patients or survivors [18,19].

The lockdown of the COVID-19 pandemic impacted PA levels around the world, ref. [20] affecting health parameters, such as cardiorespiratory fitness capacity, which is directly related to morbidity and mortality [21]. Evidence from a Canadian study suggested that the COVID-19 pandemic reduced PA levels by 33% in the YAC population, ref. [22] associating the reduction in PA levels with lower self-reported HQoL and higher fatigue levels [23]. Therefore, the COVID-19 lockdown might aggravate the common side effects of cancer therapies (i.e., fatigue or HQoL), potentially increasing the risk of CVD in this population [24]. The impact of the pandemic goes beyond traditional risk factors, as it also affects lifestyle, and familial, economic and psychosocial health. To our knowledge, there is no evidence of the impact of the lockdown on the Spanish YAC population.

The aim of this was to analyse the changes in PA levels before, during and after the COVID-19 lockdown of the YAC population in Spain, and its impact on anthropometric measures, sedentary behaviours, HQoL, and fatigue levels.

## 2. Materials and Methods

### 2.1. Study Design

The YOUNGmove study was conducted by the Faculty of Science and Physical Activity and Sport of the University of Castilla La-Mancha in Toledo. This multi-centre, retrospective, observational study utilised a web survey at three different time points between June 2020 and March 2022. The "Before lockdown" period from 14 March 2019 to 13 March 2020 (before COVID-19 restrictions) was considered. The "During lockdown" period was from 14 March 2020 to 9 May 2020 (when it was forbidden to go outside), and "after lockdown" was from 10 May 2020 (when people could go outside) to 14 March 2022 (survey closure) (Figure 1). The study was approved by the ethics committee of each hospital.

| Before lockdown | During lockdown | After lockdown |
|---|---|---|
| 14 March 2019 to 13 March | 14 March to 9 May 2020 | 10 May 2020 to 14 March |

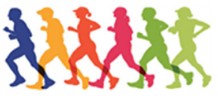 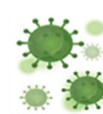 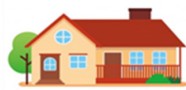 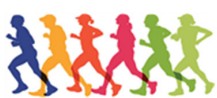 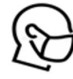

**Figure 1.** Lockdown of COVID-19 pandemic period in Spain.

*2.2. Sample*

The inclusion criteria for our study included all adults between the ages of 18–45 years, who were diagnosed with cancer before 13 March 2020; participants at any point of the cancer continuum (during treatment, in remission, or with metastatic disease); and participants who were living in Spain during the COVID-19 pandemic lockdown. Regarding the age range, the reason for extending it to 45 years was the possibility of recruiting a larger number of patients. In addition, studies specify that regardless of the limits set, the reality is that young adults are neither a paediatric nor a geriatric population, presenting a series of needs that make them unique for the role they play in society [25,26].

The exclusion criteria included a cognitive impairment that affected participation; any physical impairment or condition in which exercise was contraindicated or limited; and a cancer diagnosis more than five years before study enrolment in non-metastatic patients.

*2.3. Recruitment*

Participants were recruited from 2 hospitals in the Castilla La-Mancha region, 3 community hospitals in Madrid and the principal research site at the university. Consent was obtained through the healthcare personnel of the hospitals' oncology departments. Healthcare personnel (oncologists or/and oncology nurses) explained the study and consent to eligible participants. Figure 2 describes the two ways in which the questionnaires were shared.

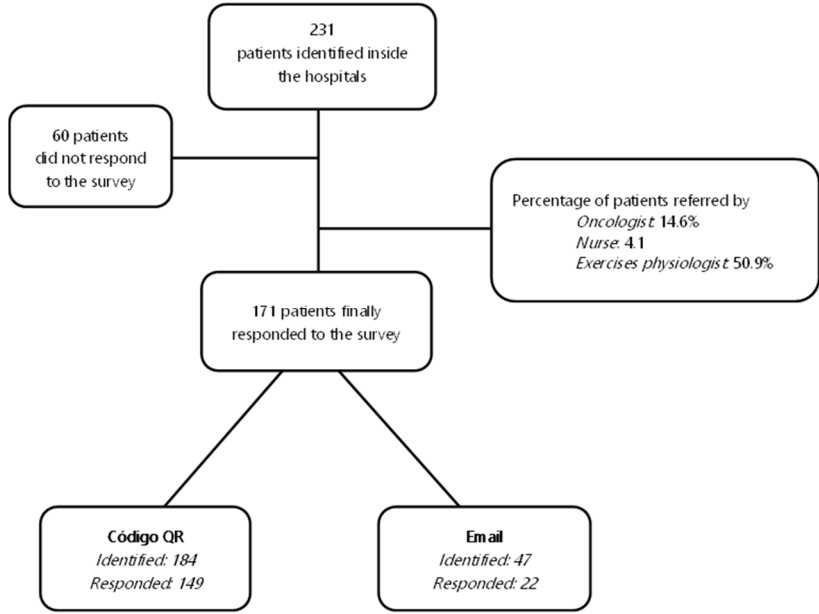

**Figure 2.** Questionnaire diffusion flow. The two ways the questionnaires were shared, based on the technological capacities of the participants. In the first pathway, patients were given an information card or other secure visual support, with which they were informed about the study objectives and accessed the questionnaire through the QR code generated. In the other pathway, the research team sent the link to the survey via email to the participants.

The YOUNGmove survey was created using the Google Form platform. Personal information was protected at all times, and the survey was delivered via an online link or QR code. Before having access to the self-reported survey, all participants provided written consent to participate in the study.

Participants completed the survey at three different time points of the COVID-19 lockdown: before, during and after. The survey included (1) a brief and clear description of the study; (2) the objectives of the study; (3) the participants' demographics including age, sex, and relevant clinical history, such as cancer diagnosis, current status, and prior and/or

current cancer therapies; (4) anthropometric measurements (i.e., weight and height); (5) and PA, HQoL and self-reported fatigue questionnaires.

## 2.4. Outcome Measures

Levels of physical activity were measured using the Spanish short version of the International Physical Activity Questionnaire (IPAQ) [27], which collected the total time of PA at various intensities per week (light, moderate and vigorous). The IPAQ asks participants to report activities performed for at least 10 min during the last 7 days. Respondents are asked to report the time spent during the physical activity performed at each of the 3 intensities: walking, moderate, and vigorous. Using the instrument's scoring protocol, total, light, moderate and high physical activity were estimated by weighing the time spent in each activity intensity with its estimated metabolic equivalent (MET) energy expenditure. The IPAQ scoring protocol assigns the following MET values to walking, moderate, and vigorous intensity activity: 3.3 METs, 4.0 METs, and 8.0 METs, respectively.

In terms of anthropometric measures, body mass index (BMI) was calculated with weight and height ($kg/m^2$), which were collected in the YOUNGmove survey.

Sedentary behaviour was measured as the total sitting time using the Spanish short version of the IPAQ, ref. [27] measured as the total sitting time per day.

HQoL was measured using the Euro-Quality of Life-5 Dimensions (EQ-5D) questionnaire, evaluating the quality of life in cancer patients considering five dimensions: mobility, self-care, usual activities, pain/discomfort, and anxiety/depression symptoms. Higher scores on the EQ-5 questionnaire indicate better health-related quality of life, with a Cronbach's alpha of 0.7 in our study [28].

Fatigue and its impact on daily activities were evaluated using the Functional Assessment of Cancer Therapy—Fatigue (FACT-F) questionnaire. This questionnaire has a total of 13 items for fatigue. Higher scores on the FACT-F questionnaire indicate lower fatigue, with a Cronbach's alpha of 0.8 in our study [29].

## 2.5. Statistical Analysis

Continuous data are reported as the mean and standard deviation (mean ± SD). Categorical data are presented as frequency and percentages. Changes between three different time points in the COVID-19 lockdown (before, during and after) and exploratory subgroup analyses from age categories were analysed using an ANOVA of repeated measures. Statistical significance was set at $p < 0.05$. For the purpose of exploratory subgroup analyses, data were further assessed using an analysis of covariance to adjust for baseline values. Subgroups included age categories (18–30, 31–40 and 40–45 years), type of tumour (breast cancer vs. other tumours) and treatment status (with active treatment vs. without active treatment). Changes in the type of tumour and treatment status were analysed using Student's *t*-test. All analyses were performed using SPSS ® V. 21.0 for Windows 7 (SPSS Inc., Chicago, IL, USA, EE.UU.).

## 3. Results

### 3.1. Sample Characteristics

A total of 171 participants were recruited. The characteristics of the participants are described in Table 1. The participants' mean age was 39 ± 5.9 years. Breast cancer and leukaemia were the most common cancer diagnoses, at 66.7% and 5.8%, respectively. A total of 56% of the participants were under treatment (including chemotherapy and radiotherapy) and over 60% of the participants had recent cancer surgery.

**Table 1.** Baseline characteristics and descriptive data of participants by age categories.

| | Total Sample N = 171 | | Age Category 18–30 N = 18 | | Age Category 31–40 N = 63 | | Age Category 40–45 N = 90 | |
|---|---|---|---|---|---|---|---|---|
| | **Female** | **Male** | **Female** | **Male** | **Female** | **Male** | **Female** | **Male** |
| **Age** | 39.2 ± 5.6 | 37.9 ± 7.6 | 26.5 ± 2.9 | 23.3 ± 3 | 36.6 ± 2.8 | 37 ± 2.9 | 43.3 ± 1.4 | 43.3 ± 1.9 |
| **Level of Physical Activity** | | | | | | | | |
| **Low levels** | | | | | | | | |
| Before COVID | 44 (25.7) | 8 (4.7) | 4 (2.7) | 1 (4.2) | 16 (10.9) | 1 (4.2) | 24 (16.3) | 6 (25) |
| During COVID | 74 (43.3) | 13 (7.6) | 5 (3.4) | 2 (8.3) | 28 (19) | 6 (25) | 41 (50.3) | 5 (20.8) |
| After COVID | 47 (27.5) | 11 (6.4) | 4 (2.7) | 2 (8.3) | 13 (8.8) | 2 (8.3) | 30 (20.4) | 7 (29.2) |
| **Moderate levels** | | | | | | | | |
| Before COVID | 54 (31.6) | 8 (4.7) | 5 (3.4) | 2 (8.3) | 21 (14.3) | 3 (12.5) | 28 (19) | 3 (12.5) |
| During COVID | 35 (20.5) | 5 (2.9) | 5 (3.4) | 1 (4.2) | 11 (7.5) | 2 (8.3) | 19 (12.9) | 2 (8.3) |
| After COVID | 44 (25.7) | 4 (2.3) | 6 (4.1) | 0 (0) | 18 (12.2) | 2 (8.3) | 20 (13.6) | 2 (8.3) |
| **Hight levels** | | | | | | | | |
| Before COVID | 49 (28.8) | 8 (4.7) | 5 (3.4) | 1 (4.2) | 18 (12.2) | 4 (16.7) | 26 (17.7) | 3 (12.5) |
| During COVID | 38 (22.2) | 6 (3.5) | 4 (2.7) | 1 (4.2) | 16 (10.9) | 0 (0) | 18 (12.2) | 5 (20.8) |
| After COVID | 56 (32.7) | 9 (5.3) | 4 (2.7) | 2 (8.3) | 24 (16.3) | 4 (16.7) | 28 (19) | 3 (12.5) |
| **Total of physical activity** | | | | | | | | |
| Before COVID | 1861.2 ± 1491.6 | 1720.4 ± 1056.7 | 2097.4 ± 2058.2 | 1759.9 ± 1274.5 | 1780.6 ± 1326.5 | 2069.3 ± 1143.2 | 1875.6 ± 1501.5 | 1474.8 ± 951.4 |
| During COVID | 1353.3 ± 1696.2 | 1281.2 ± 1418.7 | 2137.1 ± 2077.3 | 1288.5 ± 1070.9 | 1299.4 ± 1516.9 | 287.1 ± 369.9 | 1250.5 ± 1728.8 | 1941.5 ± 1616.8 |
| After COVID | 2067.4 ± 1931 | 1666.6 ± 1394.8 | 2245.4 ± 1436.3 | 1262.3 ± 1457.7 | 2049.4 ± 1470 | 1856.6 ± 1354.5 | 2048.2 ± 2133.6 | 1674.6 ± 1495.8 |
| **Light physical activity** | | | | | | | | |
| Before COVID | 754.3 ± 492.7 | 575.4 ± 433.9 | 820.3 ± 546.3 | 754.9547.3 | 770.4 ± 480.5 | 569.3 ± 463.6 | 731.1 ± 496.5 | 519.8 ± 400.1 |
| During COVID | 391.6 ± 567.3 | 418.7 ± 560 | 572.8 ± 663.7 | 148.5 ± 171.5 | 364.5 ± 546.9 | 167.1 ± 341.9 | 378.2 ± 565.2 | 676.5 ± 654.4 |
| After COVID | 724.9 ± 498.5 | 546.5 ± 428.4 | 809.7 ± 600.2 | 272.3 ± 336.9 | 793.8 ± 473.4 | 581.6 ± 328.1 | 661.3 ± 494.3 | 614.6 ± 501 |
| **Moderate physical activity** | | | | | | | | |
| Before COVID | 708.9 ± 645.2 | 662.5 ± 592.2 | 792.8 ± 729.2 | 945 ± 849.1 | 725.5 ± 649.6 | 705 ± 599.8 | 682.3 ± 633.6 | 540 ± 511 |
| During COVID | 356.7 ± 612.5 | 320 ± 490.2 | 591.4 ± 818.9 | 210 ± 344.7 | 309.8 ± 567.5 | 311.2 ± 289.5 | 347.7 ± 600.5 | 570 ± 568.3 |
| After COVID | 665.7 ± 714.9 | 562.5 ± 565 | 492.9 ± 762.7 | 450 ± 521.9 | 660 ± 665.3 | 817.5 ± 634.9 | 700.8 ± 744.2 | 430 ± 515.4 |

**Table 1.** *Cont.*

| | Total Sample N = 171 | | Age Category 18–30 N = 18 | | Age Category 31–40 N = 63 | | Age Category 40–45 N = 90 | |
|---|---|---|---|---|---|---|---|---|
| | **Female** | **Male** | **Female** | **Male** | **Female** | **Male** | **Female** | **Male** |
| **Hight physical activity** | | | | | | | | |
| Before COVID | 702.9 ± 874.4 | 755 ± 609.3 | 874.3 ± 1197.6 | 690 ± 521.9 | 632.7 ± 825.1 | 1065 ± 740.9 | 721.5 ± 849.1 | 570 ± 493.8 |
| During COVID | 635.9 ± 913.4 | 480 ± 683.4 | 1191.4 ± 1096.3 | 480 ± 678.8 | 654.5 ± 896.1 | 900 ± 254.6 | 523.1 ± 864 | 740 ± 790.8 |
| After COVID | 890.6 ± 1224.5 | 715 ± 765.1 | 1097.1 ± 1612.6 | 720 ± 831.4 | 840 ± 985.8 | 795 ± 818.8 | 889.2 ± 1308.9 | 660 ± 775.2 |
| **Secondary outcomes** | | | | | | | | |
| Weight (kg) | | | | | | | | |
| Before COVID | 64.3 ± 12.5 | 79.3 ± 12.6 | 59.9 ± 8.2 | 73.5 ± 6.8 | 63.6 ± 11.1 | 81 ± 9.9 | 65.6 ± 13.9 | 80.3 ± 15.5 |
| During COVID | 64.8 ± 12.9 | 79.3 ± 12.7 | 59.7 ± 8.1 | 72.8 ± 5.9 | 64.5 ± 12.6 | 79.6 ± 10.7 | 66.1 ± 13.8 | 81.3 ± 15.3 |
| After COVID | 64.7 ± 13.3 | 79.6 ± 12.6 | 58.7 ± 7.7 | 73.3 ± 6.6 | 64.5 ± 12.7 | 79.9 ± 10.6 | 65.9 ± 14.2 | 81.6 ± 15.1 |
| **BMI (kg/m²)** | | | | | | | | |
| Before COVID | 24.2 ± 4.3 | 24.6 ± 5.2 | 21.8 ± 2.7 | 21.3 ± 2.2 | 24.2 ± 3.5 | 25 ± 6 | 24.7 ± 4.8 | 25.5 ± 5.3 |
| During COVID | 24.2 ± 4.1 | 24.3 ± 4.8 | 22.1 ± 2.9 | 20.5 ± 1.9 | 23.9 ± 3.6 | 24.1 ± 2.8 | 24.7 ± 4.5 | 25.8 ± 5.9 |
| After COVID | 25.1 ± 4.3 | 25.3 ± 4.6 | 23.5 ± 3.2 | 20.8 ± 2.3 | 25.1 ± 4 | 24.1 ± 2.2 | 25.4 ± 4.7 | 27.6 ± 5.1 |
| **Sitting time (hours/day)** | | | | | | | | |
| Before COVID | 6.9 ± 3.6 | 6.9 ± 3.1 | 7.1 ± 3.7 | 8.2 ± 5.5 | 6.9 ± 3.3 | 7 ± 3 | 6.9 ± 3.7 | 6.5 ± 2.4 |
| During COVID | 9.4 ± 4.2 | 8.7 ± 3.6 | 11.3 ± 4.1 | 11.5 ± 5.8 | 10.2 ± 4.7 | 9.3 | 8.5 ± 3.8 | 7.4 ± 2.9 |
| After COVID | 7.2 ± 3.8 | 7.5 ± 4.5 | 9.4 ± 5.1 | 11.3 ± 10 | 7.8 ± 3.8 | 6.8 ± 3.1 | 6.5 ± 3.3 | 6.5 ± 1.8 |
| **Eq5d index value** | | | | | | | | |
| Before COVID | 0.902 ± 0.1 | 0.908 ± 0.1 | 0.903 ± 0.1 | 927 ± 0.1 | 0.915 ± 0.1 | 0.897 ± 0.2 | 0.893 ± 0.1 | 0.909 ± 0.1 |
| During COVID | 0.842 ± 0.1 | 0.884 ± 0.1 | 0.804 ± 0.2 | 0.910 ± 0.1 | 0.838 ± 0.2 | 0.876 ± 0.1 | 0.853 ± 0.1 | 0.88 ± 0.1 |
| After COVID | 0.882 ± 0.1 | 0.86 ± 0.2 | 0.835 ± 0.2 | 0.811 ± 0.3 | 0.892 ± 0.1 | 0.892 ± 0.1 | 0.883 ± 0.1 | 0.855 ± 0.1 |
| **Eq5d percentage of health (%)** | | | | | | | | |
| Before COVID | 75.9 ± 21.8 | 79 ± 18.9 | 75.2 ± 29.9 | 87.5 ± 12.6 | 78.1 ± 19.8 | 66.3 ± 24.6 | 74.6 ± 21.7 | 84.7 ± 11.9 |
| During COVID | 69.3 ± 23.1 | 67.5 ± 20.8 | 65.8 ± 30.9 | 72.5 ± 17 | 72.1 ± 22.8 | 59.4 ± 26.1 | 67.9 ± 21.9 | 71.9 ± 17.9 |
| After COVID | 72.4 ± 21.6 | 65.8 ± 25.6 | 68.4 ± 26.9 | 57.5 ± 44.3 | 75.1 ± 22.3 | 63.8 ± 25.7 | 71.3 ± 20.1 | 70 ± 19 |
| **Fatigue total** | | | | | | | | |
| Before COVID | 37.6 ± 8.9 | 35.5 ± 9.4 | 39.5 ± 6.3 | 36.8 ± 10.2 | 37.6 ± 9.5 | 38.3 ± 11.2 | 37.3 ± 0.1 | 33.3 ± 8.2 |
| During COVID | 34.5 ± 11.7 | 33 ± 13 | 31.6 ± 15.4 | 32.8 ± 23.4 | 34.1 ± 13.1 | 33 ± 12.6 | 35.3 ± 9.8 | 33.2 ± 10.2 |
| After COVID | 36.4 ± 10 | 34.3 ± 11.1 | 33.4 ± 13.2 | 33.5 ± 16.3 | 36.7 ± 10.9 | 36.5 ± 11.2 | 36.7 ± 8.6 | 33.1 ± 9.9 |

Data are presented as mean ± SD or *n* (%). Abbreviations: M = mean; SD = Standard Deviation; *N* = number of participants, % = percentage.

### 3.2. Physical Activity Levels

Descriptive data of PA are shown in Table 2. Before the lockdown, 33% of participants reported low PA levels. During the lockdown, PA levels declined as 50.9% reported low PA levels, but increased back to the levels before lockdown. Breast cancer patients reported the lowest PA levels at all three time points of the lockdown.

**Table 2.** Baseline characteristics and descriptive data of participants by type of tumour and type of patient.

|  | Breast Cancer N = 114 | Other Tumours N = 57 | With Treatment N = 75 | No Treatment N = 96 |
|---|---|---|---|---|
| **Age** | 39.9 (4.8) | 37.1 (7.3) | 38.8 (5.9) | 39.2 (5.9) |
| **Gender** |  |  |  |  |
| Female | 113 (66.08) | 34 (19.88) | 65 (38) | 82 (48) |
| Male | 1 (.58) | 23 (13.45) | 10 (5.8) | 14 (8.2) |
| **Level of physical activity** |  |  |  |  |
| **Low levels** |  |  |  |  |
| Before COVID | 35 (20.5) | 17 (9.9) | 19 (11.1) | 33 (19.3) |
| During COVID | 58 (33.9) | 29 (17) | 31 (18.1) | 56 (32.7) |
| After COVID | 38 (22.2) | 20 (11.7) | 20 (11.7) | 38 (22.2) |
| **Moderate levels** |  |  |  |  |
| Before COVID | 44 (25.7) | 22 (12.9) | 34 (19.9) | 28 (16.4) |
| During COVID | 26 (15.2) | 14 (8.2) | 19 (11.1) | 21 (12.3) |
| After COVID | 32 (18.7) | 16 (9.4) | 20 (11.7) | 28 (16.4) |
| **Hight levels** |  |  |  |  |
| Before COVID | 35 (20.5) | 22 (12.9) | 22 (12.9) | 35 (20.5) |
| During COVID | 30 (17.5) | 14 (8.2) | 25 (14.6) | 19 (11.1) |
| After COVID | 44 (25.7) | 21 (12.3) | 35 (20.5) | 30 (17.5) |
| **Total of physical activity** |  |  |  |  |
| Before COVID | 1757.8 ± 1366.6 | 2008,9 ± 1567 | 1836.4 ± 1346.3 | 1845.4 ± 1510.7 |
| During COVID | 1346.1 ± 1701.8 | 1337.1 ± 1576.7 | 1545.2 ± 1794.5 | 1185.3 ± 1531.2 |
| After COVID | 2114.5 ± 1981.5 | 1804.6 ± 1612.4 | 2298.6 ± 1929.2 | 1786.7 ± 1796.2 |
| **Light physical activity** |  |  |  |  |
| Before COVID | 742.5 ± 478.1 | 702.5 ± 509.8 | 820.4 ± 469.9 | 657.9 ± 491.9 |
| During COVID | 407.72 ± 559.8 | 370.8 ± 578.7 | 458 ± 584.6 | 346.5 ± 546.8 |
| After COVID | 730.8 ± 479.6 | 638.3 ± 514.8 | 749.8 ± 491.7 | 661 ± 491.6 |
| **Moderate physical activity** |  |  |  |  |
| Before COVID | 664.7 ± 625.1 | 777.9 ± 658.1 | 780 ± 654 | 641.9 ± 619.3 |
| During COVID | 377.4 ± 625.4 | 300 ± 532.9 | 444 ± 664.5 | 279.4 ± 528.3 |
| After COVID | 690 ± 727.4 | 573.7 ± 625.1 | 764 ± 772.6 | 563.1 ± 618.3 |
| **Hight physical activity** |  |  |  |  |
| Before COVID | 631.6 ± 808.5 | 867 ± 888.8 | 622.4 ± 809.7 | 778.8 ± 862.5 |
| During COVID | 608.42 ± 911.8 | 625.3 ± 835.6 | 726.4 ± 933.3 | 526.3 ± 839.3 |
| After COVID | 903.2 ± 1255.4 | 791.6 ± 986.1 | 1019.2 ± 1253 | 746.3 ± 1093.9 |
| **secondary outcomes** |  |  |  |  |
| **Weight (kg)** |  |  |  |  |
| Before COVID | 63.4 ± 12 | 72 ± 14.8 | 64 ± 12.4 | 68.3 ± 14.2 |
| During COVID | 64.5 ± 12.6 | 71.6 ± 15 | 64.7 ± 12.6 | 68.7 ± 14.6 |
| After COVID | 64.5 ± 13.2 | 71.4 ± 15 | 64.6 ± 12.7 | 68.5 ± 15.1 |
| **BMI (kg/m$^2$)** |  |  |  |  |
| Before COVID | 24.3 ± 4.5 | 24.3 ± 4.2 | 24.6 ± 4.5 | 24.1 ± 4.3 |
| During COVID | 24.1 ± 4.2 | 24.4 ± 4.2 | 24.3 ± 4.4 | 24.1 ± 4.1 |
| After COVID | 25.1 ± 4.5 | 25.3 ± 3.9 | 25.4 ± 5.5 | 24.9 ± 4 |

**Table 2.** *Cont.*

|  | Breast Cancer N = 114 | Other Tumours N = 57 | With Treatment N = 75 | No Treatment N = 96 |
|---|---|---|---|---|
| **Sitting time (hours/day)** |  |  |  |  |
| Before COVID | 7 ± 3.2 | 6.7 ± 4 | 6.8 ± 3.7 | 7 ± 3.8 |
| During COVID | 8.9 ± 3.7 | 9.9 ± 4.9 | 8.6 ± 3.3 | 9.9 ± 4.7 |
| After COVID | 7.1 ± 3.6 | 7.7 ± 4.5 | 7 ± 3.4 | 7.5 ± 4.2 |
| **Eq-5d index value** |  |  |  |  |
| Before COVID | 0.9 ± 0.1 | 0.9 ± 0.1 | 0.93 ± 0.1 | 0.9 ± 0.1 |
| During COVID | 0.86 ± 0.2 | 0.83 ± 0.2 | 0.9 ± 0.2 | 0.8 ± 0.2 |
| After COVID | 0.901 ± 0.1 | 0.9 ± 0.2 | 0.9 ± 0.1 | 0.9 ± 0.1 |
| **Eq-5d percentage of health (%)** |  |  |  |  |
| Before COVID | 75.5 ± 21.4 | 78.1 ± 21.5 | 74.7 ± 22.7 | 77.8 ± 20.4 |
| During COVID | 68.3 ± 23.5 | 70.5 ± 21.4 | 66.1 ± 25.5 | 71.3 ± 20.4 |
| After COVID | 72.2 ± 21.1 | 70.1 ± 24.6 | 71 ± 23 | 71.9 ± 21.8 |
| **Fatigue total** |  |  |  |  |
| Before COVID | 37.6 ± 8.9 | 36.7 ± 9.2 | 39 ± 8.2 | 35.9 ± 9.5 |
| During COVID | 35.9 ± 10.8 | 31.2 ± 13.4 | 37.3 ± 10.4 | 32 ± 12.5 |
| After COVID | 37.1 ± 9.3 | 33.9 ± 115 | 38.4 ± 10.4 | 34.3 ± 9.7 |

Data are presented as mean ± SD or *n* (%). Abbreviations: M: mean; SD: standard deviation; *n* = number of participants; % = percentage; MET = metabolic equivalent task; kg = kilograms; m$^2$ = square metre.

Table 3 identifies changes in PA levels between the time points. The amount of total PA significantly decreased from before to during the lockdown by 27.1%. There was a significant increase (49.7%) from during to after the lockdown. Similarly, significant differences were observed in the amount of light PA between before and during the lockdown, decreasing by 45.8% during confinement, though increasing significantly by 77% from during to after the lockdown (Table 3).

**Table 3.** Repeated measures ANOVA analysis examining the changes between the three time points measured (before, during and after COVID-19 lockdown).

|  | Before–During Lockdown DifM (95% CI) | *p* Value | Before–After Lockdown DifM (95% CI) | *p* Value | During–After Lockdown DifM (95% CI) | *p* Value |
|---|---|---|---|---|---|---|
| | | | *Physical activity levels* | | | |
| **Total Activity (METS)** | −498.3 (−800.97 to −195.68) | **0.0001 *** | 169.7 (−121.11 to 460.53) | 0.480 | 668.04 (384.2 to 951.88) | **0.0001 *** |
| **High (METS)** | −96.1 (−258.90 to 66.62) | 0.465 | 155.8 (−43.87 to 355.45) | 0.183 | 251.9 (75.99 to 427.87) | **0.002 *** |
| **Moderate (METS)** | −350 (−491.54 to −210.21) | **0.0001 *** | −51.2 (−182.18 to 79.72) | 1.00 | 299.7 (174.50 to 424.80) | **0.0001 *** |
| **Light (METS)** | −333.8 (−455.92 to −211.61) | **0.0001 *** | −29.2 (−119.32 to 60.85) | 1.000 | 304.5 (211.61 to 455.92) | **0.0001 *** |
| | | | *Anthropometric measures* | | | |
| **Weight (Kg)** | 0.47 (−0.12 to 1.06) | 0.169 | 0.35 (−0.42 to 1.12) | 0.809 | −0.12 (−0.55 to 0.31) | 1.00 |
| **BMI (kg/m$^2$)** | −0.08 (0.334 to −0.498) | 1.00 | 0.85 (0.27 to 1.4) | **0.002 *** | 0.93 (0.4 to 1.5) | **0.0001 *** |
| | | | *Patients reported outcomes* | | | |
| **Sitting time (hours)** | 2.4 (1.6 to 3.1) | **0.001 *** | 0.363 (−0.31 to 1.04) | 0.582 | −2 (−2.6 to −1.4) | **0.001 *** |
| **EQ5D index value** | −0.5 (−0.922 to −0.884) | **0.0001 *** | −0.02 (−0.874 to −0.823) | 0.110 | 0.03 (−0.900 to −0.858) | **0.004 *** |
| **EQ5D percentage of health (%)** | −7.4 (−3.811 to −10.9) | **0.0001 *** | −4.9 (−1.3 to −8.5) | **0.004 *** | 2.5 (5.1 to −0.14) | 0.069 |
| **Fatigue Total** | −2.9 (−4.75 to −1.24) | **0.0001 *** | −1.2 (−2.89 to 0.41) | 0.211 | 1.8 (.24 to 3.27) | **0.0017 *** |

Data are presented as mean ± SD or *n* (%). Abbreviations: DifM: mean difference; CI: confidence interval; %Dif: percentage of difference; METS: metabolic equivalent of task; kg: kilogram; m$^2$: square metre; %: percentage; * = significant different observed (*p* < 0.05).

The amount of moderate PA decreased significantly by 49% between before and during the lockdown and increased by 85.2% from during to after the lockdown. The amount of high PA only showed a significant increase between during and after the lockdown, increasing by 41%. Non-significant changes were seen before and after the lockdown in all types of PA (Table 3). Curiously, the only significant changes found between subgroups were that patients undergoing treatment had a significantly higher

level of light PA before the lockdown compared to untreated patients (DifM, 162.4 METS; $p = 0.03$; 95% IC, 15.7 to 309.2).

*3.3. Secondary Objectives Results*

3.3.1. Anthropometrics Measure

Before the lockdown, the participants' mean BMI was $24.3 \pm 4.4$ kg/m$^2$, which did not change during the lockdown ($24.2 \pm 4.2$ kg/m$^2$). Compared to before and during the lockdown, BMI significantly increased after the lockdown ($25.3 \pm 4.3$ kg/m$^2$; DifM, 0.85; $p = 0.002$; 95% CI, 0.27 to 1.4; DifM, 0.9; $p = 0.0001$; 95% CI, 0.4 to 1.5, respectively). After the lockdown, an increase in BMI was observed in the following groups: breast cancer participants ($25.1 \pm 4.5$ kg/m$^2$), participants with other tumours ($25.3 \pm 3.9$ kg/m$^2$), participants undergoing treatment ($25.4 \pm 5.5$ kg/m$^2$), age category of 31–40 ($25 \pm 3$ kg/m$^2$), and age category above 41 y ($25.7 \pm 5.7$ kg/m$^2$) (Tables 1 and 2).

3.3.2. HQoL

Participants before lockdown had an EQ-5D score of $0.9 \pm 0.12$ which is under normal values; however, it decreased significantly during the lockdown by 58.9% (Table 3). An improvement of 3.5% was reported from during to after the lockdown, but levels remained lower than the baseline (Table 3). HQoL was similar in all subgroups analysed before and after the lockdown. The percentage of self-reported health values declined by 9.7% during the lockdown compared to before (Table 3). Most participants who perceived a low percentage of health were in active treatment (Table 2); however, patients undergoing active treatment reported higher HQoL levels compared to untreated patients before the lockdown (DifM, 0.04; $p = 0.027$; 95% CI, 0.005 to 0.08), during (DifM, 0.05; $p = 0.04$; 95% CI, 0.002 to 0.1), and after the lockdown (DifM, 0.05; $p = 0.03$; 95% CI, 0.005 to 0.09).

3.3.3. Fatigue

Most participants reported low fatigue levels before the lockdown. Fatigue levels increased from before to during the lockdown, with a decrease in fatigue levels from during to after the lockdown (Table 1). The lowest fatigue levels were reported by patients undergoing treatment at the three different points of the lockdown (Table 2).

Fatigue levels increased by 8% during the lockdown compared to before. Non-significant changes were found before and after the lockdown (Table 3). Significantly lower fatigue levels were observed in patients undergoing treatment compared to untreated patients before (DifM 39.5; $p = 0.029$; 95% CI, 4.7 to 74.8), during (DifM; 5.3, $p = 0.004$; 95% CI, 1.7 to 8.8), and after the lockdown (DifM, 52.8; $p = 0.009$; 95% CI, 13.18 to 92.32).

3.3.4. Sedentary Behaviours

During the lockdown, sitting time increased by 35% compared to before the lockdown ($9.3 \pm 4.2$ vs. $6.9 \pm 3.5$ h) but improved by 21.5% from during to after the lockdown ($9.3 \pm 4.2$ vs. $7.3 \pm 3.9$ h; Table 3). However, changes between before and after the lockdown were non-significant. Non-significant changes were observed in the subgroup analysis.

## 4. Discussion

Our study showed the impact of the COVID-19 lockdown on different health-related parameters in a cohort of YAC patients in Spain. The main conclusion of this study was that a 27.1% reduction in PA levels was observed during this period, which was not fully recovered post-lockdown, particularly for high-intensity PA levels. This decline, coupled with an increase in sedentary activities, may have long-term cardiovascular consequences in this population [30,31].

In concordance with our results, a cross-sectional study reported similar findings, where vigorous and moderate PA significantly decreased in healthy Spanish adults during the pandemic [32]. Physical inactivity in cancer patients is related to increased psychosocial

and metabolic comorbidities, and we are at the beginning of understanding the full impact that the pandemic had on the cancer population in Spain.

We found that self-reported HQoL worsened, and fatigue levels increased during the lockdown. Although these parameters improved, these levels remained lower compared to pre-lockdown levels. Surprisingly, patients undergoing treatments reported higher HQoL and lower fatigue levels at all three time points. We attribute this finding to the active treatment participant's ability to leave their houses, despite the imposed restrictions during the pandemic. Travel to the hospital for treatment and increased contact with healthcare personnel may explain these results of HQoL and fatigue. In an Italian cancer cohort with active treatment, HQoL worsened during the COVID-19 lockdown [33]. Similarly to our findings, those of a French study which included cancer patients who were receiving active treatment showed low levels of fatigue during COVID-19 lockdown [34]. Based on these data, the country, type, and duration of restriction could have had an impact on health parameters of cancer patients under treatment.

We found an increase of 35% in levels of sedentarism during the lockdown, with an improvement post-lockdown. These results are in agreement with those of a previous cross-sectional study that observed an increase in sedentary time by 23.8% amongst the general Spanish population during the lockdown, with the greatest increase being 47.7% in Spanish young adults (18–24 years) [32]. An observational cross-sectional study of 600 cancer survivors reported that 44% of participants did not meet the PA guidelines' levels, with increasing levels of inactivity and sedentarism during the pandemic [23]. Evidence from prospective analyses showed that sedentary behaviour was associated with an increased risk of developing CVD and other metabolic comorbidities [35,36]. Furthermore, inactive cancer survivors (with sitting times of >8 h/d) have up to a five-fold greater risk of all-cause mortality, cancer, and cardiovascular mortality [37–39].

Physical inactivity is associated with a reduction in muscle insulin sensitivity, loss of cardiorespiratory fitness, central and peripheral cardiovascular function, skeletal muscle oxidative metabolism, and mitochondrial function [40–42]. Considering that cancer patients and survivors are at an increased risk of metabolic comorbidities [36,43], due to the direct and indirect damage of the cancer therapies, physical inactivity would further impair their health status [44,45]. A recent meta-analysis showed that reductions in PA status are related to decreased cardiorespiratory fitness capacity [45]. In cancer patients, peak VO2 is an excellent predictor of health and survival [46,47]. The use of cancer therapies has been associated with a reduction of up to 26% of the peak VO2, and patients may not recover to baseline levels after therapy completion [48,49]. It is important to emphasise that young adults can tolerate more intensive chemotherapeutic regimens compared to older adults [50]. Consequently, cancer survival rates have been increasing in YAC patients, ref. [51] but there is a higher risk of long-term side effects [52].

PA has proven to be an effective therapy to prevent and mitigate the reduction in peak VO2 associated with cancer therapies [14]. Strong evidence from randomised control trials supports the use of exercise to maintain and improve peak VO2 during and after cancer therapies, showing positive benefits for psychosocial health and reducing treatment side effects, such as fatigue and reduced HQoL [53,54]. The lockdown during the COVID-19 pandemic and its impact on PA levels could potentially result in further impairment of the cardiovascular, pulmonary, and peripheral neural systems of cancer patients and survivors [21]. Physical inactivity can also affect mitochondrial enzyme activities, modifying the metabolic phenotype toward that of a glycolytic fibre, and making the muscle more susceptible to fatigue [55,56].

In our study, the impact of participants on self-reported HQoL was of major importance, as we saw a decline of almost 60% of these levels during the lockdown. Our results show the need to implement preventive and therapeutic strategies to improve overall health across the cancer continuum. A proposed intervention is cardio-oncology rehabilitation (CORE), a program focused on improving health behaviours, risk factors, and cardiovascular outcomes in cancer patients. An exercise-based intervention can be tailored

to the uniqueness of every patient and can be delivered in-person or online, increasing the accessibility to a broad range of cancer patients [47].

For this reason, implementing accessible and proven interventions, such as CORE, provides an excellent option to maintain and improve health parameters in the YAC population. In situations such as the lockdown, we must innovate and consider how to best approach this population. Online exercise programs are a feasible and good alternative with which to address the health needs of YAC patients, including their active work life, independence, higher education, and family responsibilities [50].

*Limitations*

Our study showed the impact on health parameters and its relation to physical activity and sedentarism in a large sample of Spanish YAC participants across three different time points of the COVID-19 lockdown. Important limitations include the self-reported and retrospective nature of this study, which could result in result bias. Participants' answers may have resulted in under- or over-estimations about the information requested in this survey. That the recruitment age was extended to 45 years makes it difficult to compare this study with other studies in this population. Finally, our sample includes mostly women with breast cancer; therefore, the findings may not be widely applicable to men and other cancer populations.

**5. Conclusions**

In this Spanish cohort of YAC participants, we showed the impact of the COVID-19 pandemic on different health parameters, including a significant reduction in PA levels, increased sedentary behaviours, increased fatigue levels, and a substantial reduction in HQoL during the lockdown. Most analysed health parameters improved post-lockdown; however, self-reported fatigue and HQoL levels did not recover to baseline levels. For these reasons, we emphasise the importance of implementing strategies such as CORE, which can be delivered online, to reduce the impact of cancer therapies under conditions such as the COVID-19 lockdown.

**Author Contributions:** Conceptualisation, M.C.-M. and S.C.-B.; methodology, S.C.-B.; validation, S.L.-T., M.M.S., S.C.G., A.F.A. and A.R.-C.; formal analysis, M.C.-M.; investigation, M.C.-M.; resources, V.G.-C. and N.R.-C.; data collection, M.C.-M., S.C.-B., L.G.-H., S.L.-T., M.M.S., S.C.G., A.F.A. and A.R.-C.; writing—original draft preparation, M.C.-M.; writing—review and editing, M.C.-M., S.C.-B., F.R.-T. and R.L.; visualisation, S.C.-B., V.G.-C. and N.R.-C.; supervision, S.C.-B., F.R.-T. and R.L., V.G.-C. and N.R.-C.; project administration, M.C.-M. All authors have read and agreed to the published version of the manuscript.

**Funding:** This research received no external funding.

**Institutional Review Board Statement:** The study was conducted in accordance with the Declaration of Helsinki and approved by the ethics committees of La Mancha Centro General Hospital (date: 24 March 2021, no. 168-C) and Gregorio Marañon University Hospital (date: 14 May 2021, no. 168-C9. These ethics committees' approval was valid to other participated hospitals.

**Informed Consent Statement:** Informed consent was obtained from all subjects involved in the study through the survey.

**Data Availability Statement:** Data supporting reported results in this study can be found on the principal investigator's computer (Mónica Castellanos-Montealegre) with a password.

**Acknowledgments:** This study was part of a Ph.D. in social health and physical activity research project of Mónica Castellanos-Montealegre. We thank all the participants since otherwise it would not have been possible to carry out this study, as well as all the healthcare personnel for their hard work and ongoing support. Additionally, we thank Castilla-La Mancha University, especially The Faculty of Physical Activity and Sports Science of Toledo for making this research possible.

**Conflicts of Interest:** The authors declare no conflict of interest. The authors declare that no funds, grants, or other support were received during the preparation of this manuscript.

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
