# Peer review of "Impact of the COVID-19 Lockdown on Physical Activity Levels and Health Parameters in Young Adults with Cancer"

_curroncol, doi:10.3390/curroncol30060409_

Round 1

Reviewer 1 Report

1.      The current tables are difficult to read and require improvement and reorganization.

2.      This study measures physical activity levels and other psychosocial outcomes before, during, and after COVID. However, physical activity levels can vary during COVID due to the different stages of lockdowns and restrictions. Therefore, the authors should specify the time frame during COVID when the measurements were taken.

3.      The authors selected 18-45 as the age range for young adults with cancer. However, it is important to understand the rationale behind this choice as the usual age range for adolescents and young adults with cancer in the US is 18-39. The authors should provide a rationale for their selection of the age range.

4.      In the introduction section, the authors discuss meeting physical activity guidelines, but they do not specify which guidelines they followed, such as frequency, duration, and intensity. While the authors cite references, they should also provide more specific information in the text.

5.      The authors should provide more detailed information on the outcome measures used in the study, including Cronbach's alphas, particularly in regards to how light, moderate, and vigorous physical activity were defined and whether any cut-off points were used.

6.      In the results section, the authors state that "self-reported health-related quality of life (HQoL) in most participants was normal before the lockdown." It is important to clarify what is meant by "normal" HQoL and to provide the score range for the EQ-5D, which is used to measure HQoL. Additionally, the authors should clarify if the EQ-5D is a tool that is capable of measuring "normal" HQoL.

Reviewer 2 Report

Firstly, I appreciate the invitation to review the paper entitled "Impact of the Covid-19 Lockdown on Physical Activity Levels and Health Parameters in Young Adults with Cancer."

The research is interesting because physical activity is a fundamental element for the recovery of patients with different pathologies, in this case cancer. In order to improve the quality of the paper, I make the following suggestions to the authors.

The scientific language should be direct and clear. Therefore, eliminate the phrase (line 77) "The "Physical activity and health questionnaire in young adults with cancer" (YOUNGmove) project was born with the main..." Instead, write "The aim of this..."

In the "design" section (line 82), the objective should not be repeated and should not be included in this section. Only the study design should be described.

The information contained in lines 90-92 related to the procedure of how participants were recruited should be included in the "procedure" section.

Line 93 "sample." More information about the sample should be given. Mean age, % sex. Describe the sample and clarify if it is a convenience sample.

Statistical analysis. This section should be organized. For example, if estimates are made with repeated measures ANOVA, the variables should be grouped and not appear twice in the text.

I recommend that the authors redo the tables; the current ones are confusing and difficult to read. I suggest the following:

a) A table where only age groups disaggregated by sex are listed in the columns, and variables are listed in the rows.

b) I suggest that neither municipalities nor their characteristics, nor hospitals be included. In my opinion, this disaggregated information is not relevant to the research.

c) A table where only treatment and type of cancer are listed in the columns, and variables are listed in the rows.

A table where only "Before," "During," and "After" are listed in the columns and means and SD are listed in the rows. The p-values should be removed, and asterisks or superscripts should be used instead to indicate which groups are compared in the table footnote. For example: Note: a: Before vs During Lockdown; b: Before vs After Lockdown. Although it is logical to report the significance of the repeated measures ANOVA.

Figure 2 Boxplot refers to the median, so either the median values should be reported in the tables, and non-parametric tests should be performed, or Figure 2 should be removed.

Author Response

I attach the comments into the manuscript. 

Round 2

Reviewer 1 Report

The authors have not addressed my comments in the manuscript.

1. The authors should specify a specific time period for the "before lockdown" measurement of physical activity (PA) in their manuscript. For example, they could define it as the period from March 14th, 2019, to March 13th, 2020, covering the year prior to the lockdown period. Otherwise, participants could report their 2015 PA or something, 5 years before even 20 years before lockdown, which could potentially distort the findings of the study.

2. The authors should provide the rationale behind their choice of age range for young adults in their manuscript, as requested by the reviewer. This information should be included in the main text rather than just in the response file.

3. even though I requested the detailed information for the "Outcome Measures," they did not address them in the manuscript. If you don't have clear ideas about how it would look like, please refer to other well-written published articles.

3.1. Please address the specific numbers for the light, moderate, and vigorous PA.

3.2. What is the normal range for the EQ-5D? the Normal range does even exist? what is their chronbach's alpha in your study? Higher score indicates better QOL?

3.3 What is the cronbach's alpha for the FACT-F in your study? Higher scores indicates worse fatigue?

4. The authors should specify the meaning of the circle and star sign in Figure 2 of their manuscript. The reviewer speculates that they may represent outliers, but the authors should clarify this in the figure caption or legend.

5. The authors should remove the red underscores in Figure 3 of their manuscript. 

Author Response

I have doughs about the comments numbers: 4 and 5. For these reason, I attach the following document: - The word document with the solved comments. (in this documents I explain my doughs about comments 4 and 5)

Round 3

Reviewer 1 Report

na